# The Relationship Between Protein Fraction Contents and Immune Cells in Milk

**DOI:** 10.3390/ani15111578

**Published:** 2025-05-28

**Authors:** Haitong Wang, Xiaoli Ren, Li Liu, Zhuo Yang, Chunfang Li, Xiangnan Bao, Ayihumaer Amantuer, Peipei Wen, Dongwei Wang, Shujun Zhang

**Affiliations:** 1Frontiers Science Center for Animal Breeding and Sustainable Production, Huazhong Agricultural University, Wuhan 430070, China; htw0411@webmail.hzau.edu.cn (H.W.); renxiaoli@webmail.hzau.edu.cn (X.R.); liuli17509991115@webmail.hzau.edu.cn (L.L.); yang_zhuo@webmail.hzau.edu.cn (Z.Y.); chunfangli0521@webmail.hzau.edu.cn (C.L.); ahu-321@webmail.hzau.edu.cn (A.A.); wenpeipei@webmail.hzau.edu.cn (P.W.); wangdwei@webmail.hzau.edu.cn (D.W.); 2Key Laboratory of Agricultural Animal Genetics, Breeding and Reproduction, Ministry of Education, Huazhong Agricultural University, Wuhan 430070, China; 3Henan Dairy Herd Improvement Center, Zhengzhou 450046, China; 4Xinjiang Academy of Animal Science, Urumqi 830063, China; 5Hebei Livestock Breeding Station, Shijiazhuang 050060, China; 6National Center of Technology Innovation for Dairy Industry, Hohhot 010020, China; 18548163596@163.com

**Keywords:** cow, mammary gland health, somatic cell score (SCS), differential somatic cell count (DSCC), immune cell count, milk protein fraction

## Abstract

Mastitis, a common udder infection in dairy cows, significantly reduces milk yield and quality, impacting both farmers and consumers. To better understand how mastitis affects milk, this study analyzed milk samples from 21,388 cows across 33 farms. Using advanced technology, researchers measured key milk components, including proteins like casein and lactoglobulin, and immune-related factors. They also examined how these components change during different stages of lactation, with varying cow parities and under mastitis conditions. The study found that milk protein levels fluctuate throughout lactation, with some proteins decreasing early on and increasing later. Older cows showed changes in protein composition, with certain proteins decreasing and others increasing. Mastitis reduced milk yield, fat, and protein content while increasing immune-related proteins. The study also identified key relationships between immune cell levels and milk composition, providing insights into how mastitis alters milk quality. These findings help farmers better manage cow health, improve milk quality, and detect mastitis earlier. By understanding these changes, the dairy industry can work toward more sustainable and efficient farming practices, benefiting both producers and consumers.

## 1. Introduction

Milk is a vital source of nutrition in the human diet, rich in protein, fat, minerals, and vitamins [1,2]. Among them, proteins occupy a core position and are typically present between 3.0% and 3.5% [3,4]. Milk protein is mainly composed of casein and whey protein, which have significant differences in structure, function, and content and jointly confer the unique nutritional value and immune characteristics of milk.

Casein accounts for about 80% of the total milk protein, mainly including αs1-casein (αs1-CN), αs2-casein (αs2-CN), β-casein (β-CN), and κ-casein (κ-CN). αs1-CN is rich in phosphorus and has significant calcium-chelating ability, ensuring the effective absorption of calcium in the digestive system [5,6]. αs2-CN is rich in disulfide bonds, providing rich essential amino acids for the human body and meeting the needs of growth and the repair of tissues [7]. β-CN not only plays an important role in maintaining the colloidal stability of milk but can also be gradually degraded into bioactive peptides with various physiological functions, such as anti-oxidation, lowering blood pressure, and immune regulation during digestion [8]. κ-CN can ensure the stability of milk in storage and transportation through its hydrophilic groups on its surface while participating in the milk coagulation process under the action of rennet [9,10].

Whey protein accounts for 20% of total milk protein and mainly includes α-lactalbumin (α-LA), β-lactoglobulin (β-LG), immunoglobulin (IgG), and lactoferrin (LF). α-LA, as a part of the lactose synthase complex in the mammary gland, promotes the synthesis of lactose. In addition, α-LA is rich in tryptophan and has the potential to participate in the regulation of sleep and mood [11]. β-LG can bind and transport fatty acids, vitamins, and other small molecules, and its antibacterial activity can also play a role in inhibiting the growth and reproduction of *Staphylococcus aureus* and *Streptococcus enterica* in the occurrence of mastitis [12,13]. IgG can specifically recognize and bind pathogens and neutralize their toxicity, thereby effectively preventing and resisting infectious diseases, is abundant in colostrum, and can provide passive immunity for newborn calves [14,15]. LF can inhibit the growth and reproduction of bacteria by binding iron ions and depriving the iron elements required for bacterial growth and plays a complementary role with β-LG in the defense of mastitis against bacterial infection. It exerts antibacterial activity and has various functions, such as antiviral, antioxidant, and immunomodulatory [13,16].

The mammary gland health of dairy cows is a core element to ensure milk quality and yield. Clinical mastitis can reduce the milk yield of dairy cows by 20–50%, and although the symptoms of latent mastitis are not obvious, long-term accumulation will also lead to a gradual reduction in milk yield, bringing huge economic losses to the dairy cattle breeding industry. Mastitis can also trigger the imbalance of hormone levels in dairy cows, affecting their estrous cycle and conception rate, increasing the empty period, reducing reproductive efficiency, and further affecting the economic benefits and sustainable development of dairy cattle breeding [17,18]. In actual production, simultaneous detection of the somatic cell count (SCC) and differential somatic cell count (DSCC) can provide more comprehensive information on the mammary gland health status in dairy cows. Using a comprehensive analysis of these two indicators, breeders can more accurately determine whether cows suffer from mastitis, as well as the severity and distribution range of mastitis, to better carry out cow health management and disease prevention and control.

The SCC is commonly used as an early indicator of mastitis, and its continuous rise may mean that dairy cows are at risk of mastitis [19]. The somatic cells in milk are composed of four types of cells, namely, polymorphonuclear leukocytes (PMNs), lymphocytes (LYMs), macrophages (MACs), and epithelial cells. In healthy mammary glands, the SCC is generally low (<100,000 cells/mL), and macrophages represent the predominant cell type [20]. In infected udders, the SCC increases, and the proportion of different cells changes, with PMNs reaching up to 95% of the total SCC [21]. An SCC of 200 × 10^3^ cells/mL is considered the threshold for subclinical mastitis [22]. During statistical analysis, the SCC was converted to a somatic cell score (SCS) to ensure the data followed a normal distribution [23]. Both the SCC and the SCS are important indicators for assessing udder health and milk quality in dairy cows. However, the SCS is more suitable for genetic evaluation and long-term analysis, while the SCC is better suited for daily monitoring and rapid diagnosis.

In recent years, the DSCC has been considered a more precise indicator for detecting mastitis and is defined as the proportion of PMNs and LYMs in the SCC, while 1-DSCC represents the proportion of MACs in the SCC [24,25]. The DSCC can more accurately reflect the local condition of mastitis, while the SCC may be affected by multiple factors, such as individual differences in dairy cows, lactation stages, etc. For example, the SCC may be physiologically elevated during the late lactation phase of a cow, but this does not necessarily mean that the cow suffers from mastitis. However, the elevation of the DSCC more directly points to the development of mastitis [26]. Schwarz et al. (2020) classified udder health status into four categories based on the SCC and the DSCC: healthy (SCC ≤ 200,000 cells/mL, DSCC ≤ 65%), early mastitis (SCC ≤ 200,000 cells/mL, DSCC > 65%), mastitis (SCC > 200,000 cells/mL, DSCC > 65%), and chronic mastitis (SCC > 200,000 cells/mL, DSCC ≤ 65%) [27].

The relationship between the SCS or the DSCC and milk protein fractions determined using analytical methods [28], coagulation properties [29], and mineral content in milk [30] have been investigated; however, few studies have investigated the association between milk protein fractions predicted by external spectral models and mastitis, the SCC, and the DSCC in a large population.

Therefore, the aim of this study was to (1) gain insight into the secretion characteristics of milk protein fraction contents in milk; (2) explore the relationship between milk protein fraction contents predicted by the mid-infrared (MIR) spectral models and mastitis and immune cells; and (3) explore the strategies to improve milk quality and the feasibility of milk protein fraction content changes for characterizing mastitis in a large-scale dairy herd.

## 2. Materials and Methods

### 2.1. Data Source

Two batches of milk samples were collected from 33 dairy farms in Central China from April 2019 to December 2022. The average number of lactating cows on the farm was 650, ranging from 41 to 3527. During this period, the DHI laboratory determined the production performance data and mid-infrared spectroscopy data, such as the milk fat percentage, milk protein percentage, SCC (10^3^ cells/mL), and DSCC (%). Dataset 1 (the first batch of milk data) contained 23,253 records of 6841 lactating cows from 12 herds, ranging from parity 1–6, with a mean of 2.00 ± 1.10, and days in milk (DIM) ranging from 1 to 500, with a mean of 187.23 ± 109.55 days. Dataset 2 (the second batch of milk data) contained 46,909 DHI records of 14,547 lactating cows on 21 herds, ranging from parity 1–9, with a mean of 2.28 ± 1.32, and DIM ranging from 1 to 450 days, with a mean of 169.03 ± 106.03 days. Dataset 1 additionally contained health diagnostic records from the dairy farms. Specifically, CMT, as a rapid on-site screening tool, was performed on the four udder quarters of all dairy cows. Any cow with at least one CMT-positive quarter was classified as mastitis-positive. To enhance diagnostic accuracy, all CMT-positive cases subsequently underwent ELISA retesting to minimize false positives. Based on this, the udder health status of the cows was categorized into two groups: healthy (recorded as negative) and mastitis (recorded as positive). All cows were raised under intensive farming conditions, fed a total mixed ration (TMR), provided with free access to water, and housed in a combination of indoor barns and exercise yards.

### 2.2. Prediction of Milk Protein Fraction Contents

The deletion of missing spectra records and a total of 8 quantitative models of milk protein fractions, αs1-CN, β-CN, κ-CN, total CN, α-LA, β-LG, IgG, and LF, which were developed previously, were applied to the above 2 datasets. These eight models had an R^2^ of 0.89, 0.90, 0.62, 0.92, 0.79, 0.93, 0.89, and 0.38 and an RMSE of 0.64, 0.95, 0.53, 1.56, 0.35, 1.80, 3.03, and 0.02 in the test set, respectively. Data outside the mean ± 3 standard deviations of the predicted value were treated as outliers and eliminated.

### 2.3. Transformation of Cell Score

First, samples were analyzed for their SCC (10^3^ cells/mL) and DSCC (%) using Fossomatic™ 7DC (Foss A/S, Hiller ød, Denmark). Subsequently, based on the SCC and DSCC, the polymorphonuclear leukocyte count (PMN+LYM count, PMN+LYMC) and macrophage count (MAC count, MACC) were determined [30]:PMN+LYMC=SCC×DSCCMACC=SCC×(1−DSCC)

Finally, following the method used to convert the SCC to the SCS [23], the immune cell counts (PMN+LYMC and MACC) were transformed into immune cell scores (PMN+LYM score, PMN+LYMS; MAC score, MACS):$$PMN+LYMS={log}_{2}\left(SCC\times DSCC/1000\right)+3$$$$MACS={log}_{2}\left[SCC\times \left(1-DSCC\right)/1000\right]+3$$

### 2.4. Statistical Analysis

R software (version 4.3.2, R Foundation for Statistical Computing, https://www.r-project.org/ (accessed on 8 April 2025)) and the lme4 package version 1.1 were used to establish linear mixed models for analyzing the relationships between the milk yield, milk fat percentage, milk protein percentage, and milk protein fraction contents with the DIM, parity, mastitis status, SCS, DSCC, PMN+LYM, and MACS. Model 1 was based on Dataset 1, while models 2–4 were constructed using Dataset 2. The models are as follows:(1)$$\begin{array}{c} y_{ijopqmno}={DIM}_{i}+{Parity}_{j}+{Mastitis}_{o}+{Calvingseason}_{p}+{Samplingseason}_{q}\\ +{DIM\times SCS}_{m}+{Herd}_{n}+e_{ijopqmno} \end{array}$$
where $y_{ijopqmno}$ is the response variable (milk yield, milk fat percentage, milk protein percentage, and milk protein fraction contents); ${DIM}_{i}$ is the fixed effect of DIM class i (12 levels: <30; 31–60; 61–90; 91–120; 121–150; 151–180; 181–210; 211–240; 241–270; 271–300; 301–330; and >330); ${Parity}_{j}$ is the fixed effect of jth parity of the cow (4 levels, j = 1, parity 1, n = 9760; j = 2, parity 2, n = 6872; j = 3, parity 3, n = 4352; and j = 4, parity ≥ 4, n = 2269, and n is the sample size for the category); mastitis status was divided based on the health diagnostic records of farms$;\;{Mastitis}_{o}$ is the fixed effect of the oth health state (o = 0, healthy, n = 10,473; o = 2, diseased, n = 12,780); ${Calvingseason}_{p}$ is the fixed effect of the p calving season (p = 1, spring, n = 3835; p = 2, summer, n = 4469; p = 3, autumn, n = 7770; and p = 4, winter, n = 7179); ${Samplingseason}_{q}$ is the fixed effect of the *q* sampling season (q = 1, spring, n = 6084; q = 2, summer, n = 8903; q = 3, autumn, n = 5146; and q = 4, winter, n = 3120);${\;DIM\times SCS}_{m}$ is the interaction effect of the DIM and the SCS; ${Herd}_{n}$ is the random effect of the m herd (21 levels) and; $e_{ijopqmno}$ is the random residual:(2)$$y_{ijmn}={DIM}_{i}+{Parity}_{j}+{Herd}_{m}+e_{ijmn}$$
where $y_{ijmn}$ is the response variable; $and\;{DIM}_{i},$ ${Parity}_{j}$, ${Herd}_{m}$ are defined as in model 1:(3)$$y_{ijklmn}={DIM}_{i}+{Parity}_{j}+{SCS}_{k}+{DSCC}_{l}+{Herd}_{m}+e_{ijklmn}$$
where $y_{ijklmn}$ is the response variable;${\;SCS}_{k}$ is the fixed effect of the SCS class k (categorized into quartiles: 4 levels: k = 1, 0.06 ≤ SCS < 0.79, n = 11,806; k = 2, 0.79 ≤ SCS < 3.96, n = 11,679; k = 3, 3.96 ≤ SCS < 5.22; n = 11,707; and k = 4, 5.22 ≤ SCS ≤ 9.64, n = 11,717); and ${DSCC}_{l}$ is the fixed effect of DSCC class l (categorized into quartiles: 4 levels: l = 1, 0 ≤ DSCC < 30.05, n = 12,181; l = 2, 30.05 ≤ DSCC < 69.65, n = 11,178; l = 3, 69.65 ≤ DSCC < 84.05; n = 11,754; and l = 4, 84.05 ≤ DSCC ≤ 96.40, n = 11,796). The other factors are defined as in model 1:(4)$$y_{ijklmn}={DIM}_{i}+{Parity}_{j}+{PMN+LYMS}_{k}+{MACS}_{l}+{Herd}_{m}+e_{ijklmn}$$
where ${PMN+LYMS}_{k}$ is the fixed effect of PMN+LYMS class k (categorized into quartiles: 4 levels: k = 1, 0.54 ≤ PMN+LYMS < 3.16, n = 8718; k = 2, 3.16 ≤ PMN+LYMS < 3.78, n = 8717; k = 3, 3.78 ≤ PMN+LYMS < 5.30; n = 8718; and k = 4, 5.30 ≤ PMN+LYMS ≤ 9.52, n = 8715); and ${MACS}_{l}$ is the fixed effect of MACS class l (categorized into quartiles: 4 levels: l = 1, −1.80 ≤ MACS < 1.23, n = 8718; l = 2, 1.23 ≤ MACS < 2.85, n = 8717; l = 3, 2.85 ≤ MACS < 3.74, n = 8718; and l = 4, 3.74 ≤ MACS ≤ 8.77, n = 8715). The other factors are defined in model 2.

## 3. Results

### 3.1. Physiological Secretion Characteristics of Milk Protein Fraction Content

The milk yield, milk fat percentage, and milk protein percentage varied with the DIM and parity (Figure 1A,B,E,F), and this study analyzed changes in milk protein fraction content in large-scale dairy cows.

#### 3.1.1. Secretion Profile of Milk Protein Fraction Content with Lactation

In casein fractions, the changing profile of casein content during lactation was overall similar to that of the milk protein percentage (Figure 1B,C). As a whole, at the early stage of lactation, due to the dilution effect of the high milk yield at the peak of lactation, the total casein and the contents of each fraction decreased and then gradually increased. Specifically, the total CN content was relatively high at the early stage of lactation (23.04 ± 0.13 g/L), decreased by about 2.79 g/L at the peak of lactation, then increased steadily by about 1.54 g/L at the middle and late stages, and slightly decreased at over day 285. The contents of αs1-CN, β-CN, and κ-CN, the main components of casein, changed more consistently with a high level at early lactation (6.98 ± 0.04, 9.53 ± 0.08, and 4.05 ± 0.03 g/L, respectively), decreased to the lowest at the peak lactation (5.75 ± 0.04, 8.87 ± 0.07, and 3.58 ± 0.03 g/L, respectively), and then increased steadily to a higher level at themed-to-late lactation (6.15 ± 0.04, 9.75 ± 0.07, and 3.94 ± 0.03 g/L, respectively).

In the whey protein fractions, α-LA, β-LG, IgG, and LF showed different dynamic changes during lactation. The contents of α-LA, β-LG, and LF were all higher in early lactation, with the lowest value at the peak of lactation, and then increased with the increase in lactation period (from 0.64 ± 0.02, 7.59 ± 0.10, and 0.16 ± 0.00 g/L to 0.59 ± 0.01, 7.43 ± 0.10, and 0.15 ± 0.00 g/L and to 0.65 ± 0.01, 8.62 ± 0.10, and 0.20 ± 0.00 g/L, respectively). The IgG content was highest at the early stage of lactation (32.50 ± 0.17 g/L), which may be related to the high concentration of immune factors in the colostrum, followed by a gradual decrease, to the lowest at mid-to-late lactation (28.86 ± 0.16 g/L), and slightly increased at the late stage (30.18 ± 0.16 g/L) (Figure 1D). These changes reveal the phased pattern of the mammary gland in secreting protein and regulating immune function throughout lactation.

#### 3.1.2. Secretion Profile of Milk Protein Fraction Content with Parity

Parity not only significantly affects the milk yield, milk fat percentage, and milk protein percentage but also plays an important role in the milk fraction content. The total casein content decreased significantly with parity; the β-CN content did not differ significantly between the first three parities and decreased significantly at ≥4 parity; the κ-CN content persistently increased with parity (3.73 ± 0.03–3.82 ± 0.03 g/L); and the αs1-CN content did not change significantly with parity (5.98 ± 0.04–6.00 ± 0.04 g/L) (Figure 1G).

Whey protein showed characteristic changes with increasing parity. The α-LA content decreased with increasing parity from 0.67 ± 0.01 g/L in the first parity to 0.62 ± 0.01 g/L; the β-LG and IgG content gradually increased in the first three parities and stabilized at the third parity (8.06 ± 0.10 and 29.63 ± 0.16–29.74 ± 0.16 g/L, respectively); and the LF content increased with increasing parity from 0.15 ± 0.00 g/L at parity 1 to 0.17 ± 0.00 g/L at ≥4 parity (Figure 1H). The characteristics of changes in the casein fraction and whey protein fraction reflect the enhancement of nutritional function and immune regulation in the lactation of high-parity dairy cows.

### 3.2. Association of Milk Protein Fraction Content with Mastitis and SCS

#### 3.2.1. Differences in Milk Protein Fraction Content Under Different Mammary Gland Health Conditions

Mastitis significantly affects the milk yield, milk fat percentage, and milk protein percentage [31]. In this study, we found that the milk protein fraction content was also affected by mastitis. The milk yield, milk fat percentage, and milk protein percentage were significantly higher in the healthy group (31.53 ± 0.46 kg/d, 4.30 ± 0.03%, and 3.39 ± 0.01%) than in the diseased group (29.18 ± 0.45 kg/d, 4.08 ± 0.02%, and 3.35 ± 0.01%) (*p* < 0.05, Figure 2A,B). The contents of total casein, β-CN, κ-CN, and α-LA in the healthy group were also significantly higher than those in the diseased group (*p* < 0.05) (Figure 2C,D), which are the main nutritional proteins in milk, and their decreased contents reflect the negative impact of mastitis on the nutritional value of milk. Conversely, the contents of β-LG, IgG, and LF in the diseased group were significantly higher than those in the healthy group (*p* < 0.05) (Figure 2D), which suggests that mastitis-induced inflammatory responses may enhance the secretion of immune-related whey protein, thus playing a role in immune regulation under pathological conditions.

In summary, mastitis not only significantly reduced milk production capacity and the content of major nutritional protein in dairy cows but was also accompanied by a significant increase in immune-related protein. This result suggests that the mammary gland health status is an important factor affecting milk quality and reflects the dual effects of mastitis on mammary gland function and milk quality in dairy cows.

#### 3.2.2. Variation in Milk Protein Fraction Content with SCS

The somatic cell score (SCS) significantly affected the milk yield, milk fat percentage, milk protein percentage, and milk protein fraction content of dairy cows, showing a certain change rule. As the SCS increased from 0.06 to 9.46 (i.e., the SCC increased from 13.03 × 10^3^ cells/mL to 8803.47 × 10^3^ cells/mL), the average daily milk yield of cows decreased from 37.04 ± 0.67 kg/d to 28.66 ± 0.56 kg/d, showing a gradual downward trend (Figure 3A). At the same time, the increase in SCS was accompanied by increases in the milk fat percentage and milk protein percentage (Figure 3B), which may be related to a decrease in the proportion of water in the milk and a component concentration effect following mastitis or impaired mammary gland function. Studies have shown that the assessment of the SCC in individual milk is a valuable approach for the early detection and effective management of mastitis [32]. Total casein, αs1-CN, and β-CN contents gradually increased as the SCS increased in the range of 0.06–5.22 (an SCC of 13.03 × 10^3^–465.89 × 10^3^ cells/mL); when the SCS was higher than 5.22, all three contents significantly decreased (Figure 3C). It was speculated that three types of casein have the potential to be indicator proteins in the development of cow mammary glands from a healthy state to a subclinical mastitis state.

In whey protein, when the SCS increased in the range of 0.79–5.22 (SCC 21.61 × 10^3^–465.89 × 10^3^ cells/mL), none of the four whey protein contents changed significantly. When the SCS increased from 0.06 to 9.64, the α-LA content decreased from 0.70 ± 0.02 g/L to 0.59 ± 0.02 g/L, while the β-LG, IgG, and LF contents increased from 7.07 ± 0.12, 28.61 ± 0.2, and 0.13 ± 0.00 g/L to 8.38 ± 0.09, 29.85 ± 0.16, and 0.19 ± 0.00 g/L, respectively (Figure 3D).

In summary, the SCS and mammary inflammation affect milk quality and mammary immune function. However, the relationship between milk protein fractions and refined immune cell changes in the SCC is not clear.

### 3.3. Characteristics of Changes in Milk Protein Fraction Content with DSCCs

Differential somatic cell counts (DSCCs) reflect the proportion of PMNs, LYMs, and MACs in the overall count of somatic cells.

As shown in Figure 4, when the DCSS was <50%, i.e., the MACs in somatic cells were dominant, the milk yield decreased, and the milk protein percentage and κ-CN content increased significantly (*p* < 0.05), and the milk fat percentage and other milk protein fraction contents did not change significantly (*p* > 0.05) as the DSCC increased.

When the DSCC was >50%, that is, the PMNs and LYMs in somatic cells were dominant, the milk yield was higher, and the milk fat percentage and milk protein percentage significantly decreased with the increasing DSCC (*p* < 0.05). Total casein, αs1-CN, β-CN, and κ-CN decreased from 21.33 ± 0.13, 6.10 ± 0.04, 9.42 ± 0.07, and 3.90 ± 0.03 g/L to 21.20 ± 0.13, 5.77 ± 0.04, 9.20 ± 0.07, and 3.67 ± 0.03 g/L, respectively (*p* < 0.05); the α-LA content increased from 0.61 ± 0.02 g/L to 0.70 ± 0.02 g/L; and β-LG, IgG, and LF decreased from 8.30 ± 0.09, 30.26 ± 0.16, and 0.18 ± 0.00 g/L to 7.28 ± 0.09, 28.38 ± 0.16, and 0.13 ± 0.00 g/L, respectively (*p* < 0.05).

### 3.4. Characteristics of Changes in Milk Protein Fraction Content with Immune Cell Score

#### 3.4.1. Variation in Milk Protein Fraction Content with PMN + LYM Score

As the PMN + LYMS increased from 0.54 to 3.78 (PMN + LYMC from 18.17 × 10^3^ to 171.71 × 10^3^ cells/mL), the milk yield, milk fat percentage, and milk protein percentage did not change significantly, the PMN + LYMS increased from 0.54 to 9.52 (PMN + LYMC from 18.17 × 10^3^ to 9177.37 × 10^3^ cells/mL), the milk yield increased from 28.96 ± 0.55 kg/d to 34.85 ± 0.54 kg/d, and the milk fat percentage and milk protein percentage decreased from 4.16 ± 0.08 and 3.41 ± 0.02% to 3.89 ± 0.08 and 3.26 ± 0.02%, respectively (Figure 5A,B).

In the casein fraction, there was no significant change in the total casein and κ-CN content when the PMN + LYMS increased in the range of 0.54–5.30 (PMN + LYMC 18.17 × 10^3^–492.46 × 10^3^ cells/mL). When the PMN + LYMS continued to increase, both the total casein and κ-CN contents decreased significantly from 21.38 ± 0.13 and 3.84 ± 0.03 g/L to 21.03 ± 0.13 and 3.68 ± 0.03 g/L, respectively. αs1-CN and β-CN did not change significantly at a PMN + LYMS of 0.54–3.78 and gradually decreased from 6.09 ± 0.04 and 9.47 ± 0.08 g/L to 5.75 ± 0.04 and 9.18 ± 0.08 g/L, respectively, with increasing PMN + LYMSs (Figure 5C).

In whey protein, when PMN + LYMS ranging from 0.54 to 5.30, the α-LA and β-LG contents did not change significantly; IgG decreased slightly (from 29.81 ± 0.18 g/L to 29.54 ± 0.17 g/L); and the LF content was more stable (approximately 0.17 ± 0.00 g/L). When the PMN + LYMS increased from 5.30 to 9.52, α-LA increased from 0.63 ± 0.01 g/L to 0.67 ± 0.02 g/L, and β-LG, IgG, and LF decreased from 8.05 ± 0.09, 29.54 ± 0.17, and 0.17 ± 0.00 g/L to 7.59 ± 0.09, 28.61 ± 0.18, and 0.15 ± 0.00 g/L, respectively (Figure 5D).

Overall, all milk protein fractions decreased with increasing PMN + LYMSs except α-LA, a characteristic opposite to the SCS.

#### 3.4.2. Variation in Milk Protein Fraction Content with MAC Score

The milk yield, milk fat percentage, milk protein percentage, and milk protein fraction content changed with the MACS in a profile opposite to the PMN + LYMS (Figure 5E–H) and consistent with the SCS.

## 4. Discussion

DIM and parity significantly affect the milk yield, milk fat percentage, and milk protein percentage [33,34] and significantly affect the milk protein fraction content.

β-CN, κ-CN, total CN, and α-LA contents were significantly decreased when mastitis developed or when the SCC was >465.89 × 10 ^3^ cells/mL (Figure 2 and Figure 3). Studies have shown that milk fat, milk protein, lactose, and total casein content are all significantly reduced when mastitis occurs [35]. Caseins (β-CN, κ-CN, and total CN) and α-LA are nutritional proteins synthesized and secreted by mammary epithelial cells. When mastitis occurs, on the one hand, plasminogen is catalyzed by a plasmin plasminogen activator secreted by somatic cells to plasmin at an increased rate, plasmin binds to casein micelle substrates, and preferentially hydrolyzes β-CN into γ-casein, and small peptides are released into the whey [36,37], resulting in decreased β-CN and total CN contents; on the other hand, mammary epithelial cells are damaged due to inflammation and pathogen invasion, resulting in decreased protein synthesis capacity. At the same time, toxins released by bacteria destroy the integrity of mammary epithelial cells, blood and interstitial fluid penetrate into the mammary lumen, and the dilution effect further reduces the concentration of nutritional protein [38,39]. The decreased calcium content caused by osmotic pressure leads to decreased casein micelle stability and is more likely to cause casein hydrolysis [40]. A key protein in lactose synthesis is α-LA, and decreased lactose content is associated with the development of mastitis [29,41,42]. Therefore, to ensure milk quality, measures should be taken to strengthen mammary gland health management. For example, regular monitoring of SCCs in dairy cows can detect early signs of mastitis promptly and reduce the risk of mastitis by improving the feeding environment, optimizing feed nutritional formula, and reducing stress factors. The continuous monitoring of changes in the milk protein fraction contents can provide data support for the long-term improvement of milk quality. Optimizing mastitis prevention and control requires rational antibiotic therapy combined with non-antibiotic strategies, such as probiotics and immunopotentiators. Through the above measures, the damage of inflammation to mammary epithelial cells may be reduced and the normal secretory function of the mammary gland may be restored.

Effectively reducing the effect of mastitis on milk quality and improving the lactation efficiency and milk composition quality of dairy cows provide a guarantee for the sustainable development of the dairy industry. At the same time, these studies provide an important scientific basis for the further exploration of the potential mechanisms of mastitis prevention and control and milk quality improvement.

β-LG, IgG, and LF are immune-related proteins that are significantly increased in mastitis or high SCCs (Figure 2 and Figure 3). When the blood–milk barrier function is weakened, high concentrations of immune-related protein in serum (e.g., IgG and LF) penetrate the mammary gland follicle [39]. IgG is involved in pathogen clearance by neutralizing pathogen toxins and activating the complement system, and LF limits pathogen growth by binding iron. The secretion of β-LG is an important manifestation of the local response to inflammation in the mammary gland, and its increased levels reflect the direct immune response of the mammary gland to pathogens [28,43,44]. In mastitis or a high SCC state, a decrease in the nutrition-related protein content and an increase in the immune-related protein content in milk are the comprehensive embodiments of impaired mammary gland function and enhanced immune response caused by mastitis, reflecting the functional transition of the mammary gland from nutritional secretion to immune protection. Therefore, β-CN, κ-CN, total CN, α-LA, β-LG, IgG, and LF in milk have the potential as indicator proteins for mastitis development. The milk protein fraction content changed with PMN + LYMS in contrast to the SCS and with the MACS in the same manner as the SCS. Based on the above studies, it appears that the MACS more closely reflects the occurrence of mastitis [28,29]. Previous studies have shown that an increase in the SCS is associated with the occurrence of mastitis, and an increase in the DSCC is accompanied by the occurrence of mastitis, while the change characteristics of the milk protein fraction contents with the PMN + LYMS in this study are different from those with the SCS, indicating that the relationship between DSCCs and SCCs is nonlinear, and high levels of DSCCs or PMN + LYMSs do not represent mastitis occurrence [30]. It may be that the majority of the data used in this study came from healthy cows, and the proportion of LYMs in the DSCC may have prevailed over the PMNs. If the relationship between the milk protein fraction contents and PMN + LYMS changes is precisely assessed, PMNs and LYMs should be separated and analyzed separately.

Mastitis is the most important disease of dairy cows and causes a huge economic burden to the cattle industry. E. Rollin et al. (2015) [45] estimated direct and indirect costs per case of clinical mastitis occurring during the first 30 days of lactation by a deterministic partial budget model. An average case of clinical mastitis resulted in a total economic cost of USD 444, including USD 128 for diagnostics, therapeutics, and non-saleable milk, and USD 316 for veterinary service, labor, and death loss. For large dairy farms with a clinical mastitis prevalence rate of 0.60–18.2%, the comprehensive economic loss was about USD 12,000 to USD 76,000 per farm per month [46]. Petzer et al. (2017) [47] developed an udder health model using current data to predict future outcomes to assist in optimizing decisions and balancing the costs/benefits. Financially and epidemiologically, the best-case scenario is to create a low TR environment in *S. aureus*-positive herds. As a result, the focus of mastitis prevention should be on excellent management rather than treatment [47]. This study reveals the relationship between milk protein fraction components and mastitis and immune cells. Under the trend of intensive breeding, applying this result to the daily management of the DHI system can effectively reduce the occurrence of mastitis caused by management, thereby achieving early detection and early treatment, reducing the waste of labor and financial resources, as well as drug pollution to the environment, and achieving the healthy and sustainable development of dairy farming. Ultimately, the healthy and sustainable development of dairy farming can be achieved.

## 5. Conclusions

The results show that the content of milk protein fractions was significantly correlated with the lactation days, parity, mammary gland health, SCS, PMN + LYMS, and MACS. As the SCS increased and mastitis developed, the milk’s β-CN, κ-CN, total casein, and α-LA contents decreased, and the β-LG, IgG, and LF contents increased. When the PMN + LYMS increased, α-LA increased, and αs1-CN, β-CN, κ-CN, total CN, β-LG, IgG, and LF decreased. When the MACS increased, the milk protein fraction content changed inversely. Prevention and timely diagnostic and therapeutic measures should be taken to cope with the negative impact of mastitis and high SCSs, PMN + LYMSs, and MACSs on milk quality. Whether the milk protein fraction can be used as a predictor of mastitis in dairy cows deserves further investigation.

In addition, there is a nonlinear relationship between the DSCC and milk protein fraction content, and in order to more accurately assess the relationship between milk protein fraction content and immune cells, PMNs and LYMs need to be counted separately.

## Figures and Tables

**Figure 1 animals-15-01578-f001:**
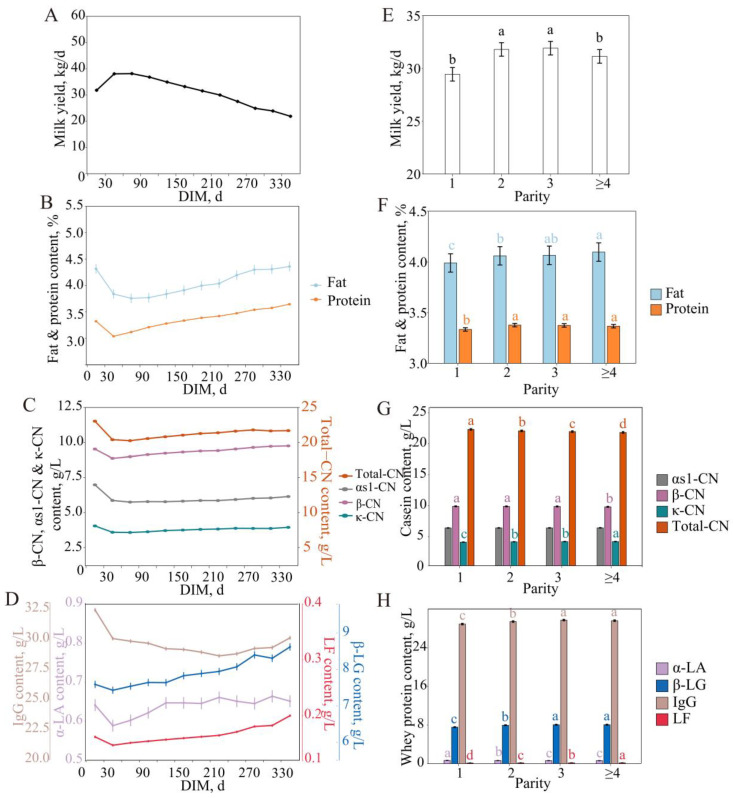
Least squares means and SE of milk yield (**A**,**E**), fat and protein (**B**,**F**), casein (**C**,**G**), and whey protein (**D**,**H**) milk content across lactation (**left**) and parity (**right**). Different lowercase letters of the same color indicated significant differences between classes at a significant level of 0.05.

**Figure 2 animals-15-01578-f002:**
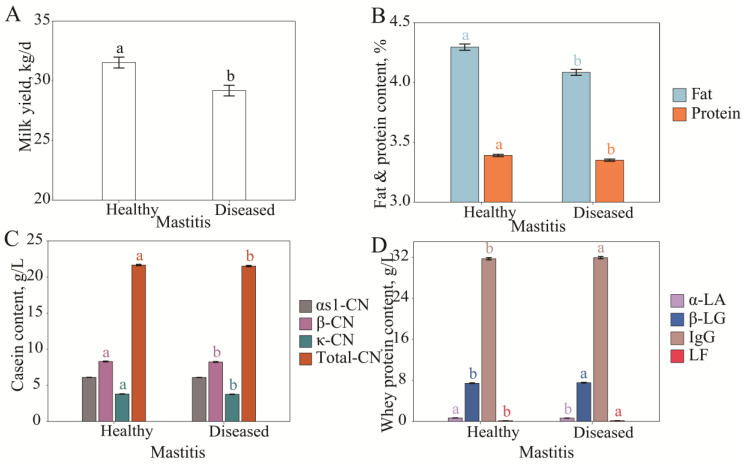
Least squares means and SE of milk yield (**A**), fat and protein (**B**), casein (**C**), and whey protein (**D**) milk content across mastitis progress classified by clinical disease records. Different lowercase letters of the same color indicated significant differences between classes at a significant level of 0.05.

**Figure 3 animals-15-01578-f003:**
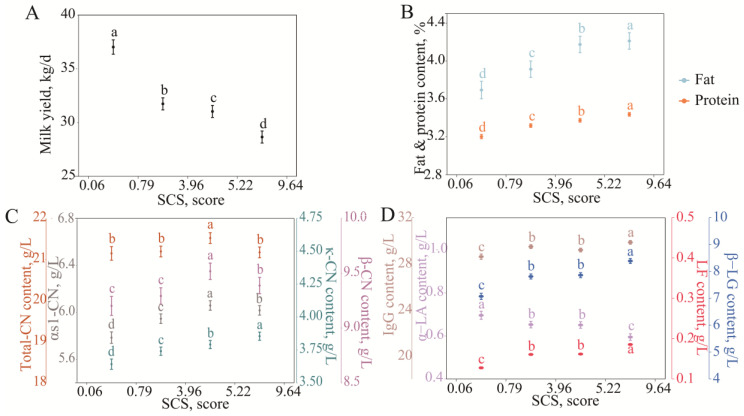
Least squares means and SE of milk yield (**A**), fat and protein (**B**), casein (**C**), and whey protein (**D**) milk content of SCS classes SCS was calculated by SCS = log_2_ [SCC/1000] + 3. Different lowercase letters of the same color indicated significant differences between classes at a significant level of 0.05.

**Figure 4 animals-15-01578-f004:**
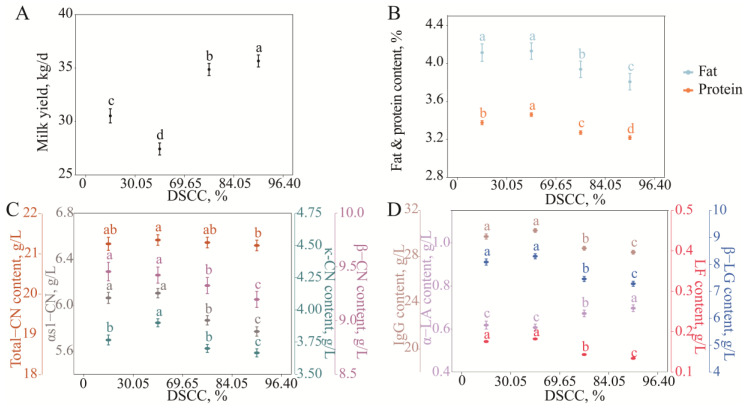
Least squares means and SE of milk yield (**A**), fat and protein (**B**), casein (**C**), and whey protein (**D**) milk content across DSCC classes. Different lowercase letters of the same color indicated significant differences between classes at a significant level of 0.05.

**Figure 5 animals-15-01578-f005:**
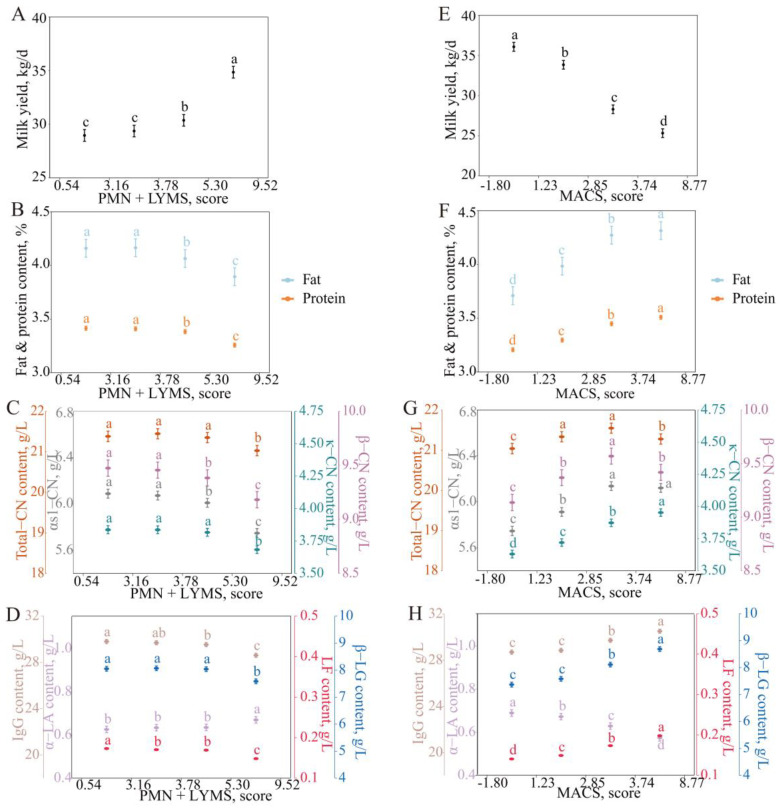
Least squares means and SE of milk yield across (**A**,**E**), fat and protein (**B**,**F**), casein (**C**,**G**), and whey protein (**D**,**H**) milk content PMN + LYMS (**left**) and MACS (**right**) classes. PMN + LYMS calculated by PMN + LYMS = log _2_ [SCC × DCSS/1000] + 3, and MACS was calculated by MACS = log_2_ [SCC × (1 − DSCC)/1000] + 3. Different lowercase letters of the same color indicated significant differences between classes at a significant level of 0.05.

## Data Availability

The data presented in this study are available upon request from the corresponding author due to data ownership reasons.

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
