# Peer review of "The Relationship Between Protein Fraction Contents and Immune Cells in Milk"

_animals, 2025, doi:10.3390/ani15111578_

Round 1

Reviewer 1 Report

Comments and Suggestions for Authors

The study attempts to examine  the secretion characteris tics of milk protein fraction contents in milk, explores the relationship between milk protein fraction contents and mastitis and immune cells, as well as the strategies to improve milk quality and the feasibility of milk protein fraction contents changes for characterizing mastitis in a large-scale dairy herd.

The information is not new and there is a lot of knowledge on the subject in the literature.

The study included many cows, of different lactation and herds, and measured many components in the milk  include SCC and DSCC. However, the analysis of the data is simplistic.

First of all, it is not clear to which cows they are referring, to healthy cows or mastitis cows according to the farmer's data or  to SCC and DSCC. I understand that a cow was measured once, including all the quarters without  testing how many inflamed quarters/cow, bacteria in a quarter, etc,

Second, the interactions between dependent variables  (DIM, parity and calving seasson ….) and SCC are unclear… Third, there is no data on the number of cows in each subgroup.

Regarding the statistical model: it is preferable to analyze with days from calving as a continuous variable including DIM and not as categorical.

When you want to draw your lactation curve you need to use DIM as a continuous variable including DIMXDIM and  DIM in the model.

The introduction of the cow in the model nests in a group, milking... For a more accurate calculation of the error.

Also, please explore the strategies to improve milk quality. This is one of the goals you have set.

I suggest the authors  read the publication of Leitner, Salnikov and Marin.

Please remove all (SCC)  apart from the first time change to SCC

Lines 113-116: References. There are many publications that show much lower value  of leukocytes account in milk without inflammation ~50% and ~50% epithelial cells

Lines 123-125: See studies separating epithelial cells from microphases basis of CD14 receptor

[116 somatic cell count (SCC); 130 SCC (somatic cell count) and  131 DSCC (differential somatic cell count)]

Lines 152  and 155. Please explain the numbers   23, 253 records of 6,841 lactating cows

Lines 194.  Please explain “divided based on health diagnostic records of farms”

Health cow by the farmer with 350,000 SCC ????

Lines 324 and 332. When DSCC was > 50%, that is, PMN and LYM in somatic cells were dominant, milk  yield significantly increased,

Check that it doesn't make sense.  That in an inflammatory state the cow will give more milk ????

Figure 1: please use Bar Graph when presenting data for different parity. This is  average number not continuous.

Please provide SE during the text near each average number.

Please submit separate graphs for all components for healthy and masitic cows

Author Response

Comments 1: First of all, it is not clear to which cows they are referring, to healthy cows or mastitis cows according to the farmer's data or to SCC and DSCC. I understand that a cow was measured once, including all the quarters without testing how many inflamed quarters/cow, bacteria in a quarter, etc,

Response 1:

Thank you for pointing this out. We agree with this comment.

In this study, we determined mastitis status using the California Mastitis Test (CMT) followed by ELISA confirmation, rather than relying on somatic cell count (SCC) or differential somatic cell count (DSCC) of mixed milk sample.

Specifically, CMT, as a rapid on-site screening tool, was performed on the four udder quarters of all dairy cows. Any cow with at least one CMT-positive quarter was classified as mastitis-positive. To enhance diagnostic accuracy, all CMT-positive cases subsequently underwent ELISA retesting to minimize false positives.

Our study focused on determining whether a cow had mastitis rather than analyzing pathology details at the level of individual udder quarters. And due to practical constraints (including sampling time and workforce availability), we did not systematically record the exact locations of inflamed quarters or bacterial load quantification for most positive cases. This approach aligns with our research objective of identifying mastitis-positive cows for population health management, rather than investigating localized udder pathology. In future research, we will consider detailed recording.

In response to your comments, we have revised the manuscript. Specifically, the revised content on page 4 (line 159~165) has been highlighted with red text and yellow shading:

Specifically, CMT, as a rapid on-site screening tool, was performed on the four udder quarters of all dairy cows. Any cow with at least one CMT-positive quarter was classified as mastitis-positive. To enhance diagnostic accuracy, all CMT-positive cases subsequently underwent ELISA retesting to minimize false positives. Based on this, the udder health status of the cows was categorized into two groups: healthy (recorded as negative) and mastitis (recorded as positive).

Comments 2: Second, the interactions between dependent variables (DIM, parity and calving season ….) and SCC are unclear…

Response 2:

We sincerely appreciate your insightful suggestion regarding potential interactions among predictors. In response to your comments, we conducted an anova analysis by adding the interaction effect to the original model, and tested the significance of the interaction between DIM, parity, calving season and test season and SCS in model 1. The results showed that DIM* SCS had significant effects on total-CN, α-LA and LF (p < 0.05); and the other three interactions had no significant effects on the dependent variables.

Therefore, model 1 was updated, and the results were re-statisticed and compared, and Figure 2 was re-drawn. In the future, the impact of the interaction effect will also be considered when using the model. The changes in the manuscript have been marked in red font and yellow background (line 194,205 and 289).

Comments 3: Third, there is no data on the number of cows in each subgroup.

Response 3: We apologize for the lack of number of cows in each subgroup. For parity, mastitis, calving season and sampling season in model1, these information were updated in manuscript from line 198 to 205 (red text and yellow background):
 is the fixed effect of parity j of the cow (4levels,j=1, parity 1, n= 9,760; j=2,parity 2, n= 6,872; j=3, parity3, n=4,352; j=4, parity ≥4, n=2,269); mastitis status was divided based on health diagnostic records of farms,  is the fixed effect of the o th health state (o = 0, healthy, n=10,473; o = 2, diseased, n=12,780);  is the fixed effect of the p calving season (p = 1, spring, n=3,835; p = 2, summer, n=4,469; p = 3, autumn, n=7,770; p = 4, winter, n=7179);  is the fixed effect of the q sampling season (q = 1, spring, n=6,084; q = 2, summer, n=8,903; q = 3, autumn, n=5,146; q = 4, winter, n=3120);”

Comments 4: Regarding the statistical model: it is preferable to analyze with days from calving as a continuous variable including DIM and not as categorical.

When you want to draw your lactation curve you need to use DIM as a continuous variable including DIMXDIM and DIM in the model.

The introduction of the cow in the model nests in a group, milking... For a more accurate calculation of the error.

Response 4: Thank you for your suggestion. According to the rationale of Pegolo et al. (2021) and Bisutti et al. (2022), we did not assume any linear relationship between predictors and response variables. Therefore, we discretized the explanatory variables (i.e., DIM, Parity, SCC, DSCC, PMN-LYMS, and MACS……) and created classes based on different rule. This approach was adopted to assess the pattern of explanatory variable effects. When the focus of the study is on fitting the lactation curve, we would include DIM as a continuous variable and include factors such as the cow in the model nests in a group, milking....

Pegolo, S., D. Giannuzzi, V. Bisutti, R. Tessari, M. Gelain, L. Gallo, S. Schiavon, F. Tagliapietra, E. Trevisi, P. Ajmone Marsan, G. Bit tante, and A. Cecchinato. 2021. Associations between differential somatic cell count and milk yield, quality, and technological char acteristics in Holstein cows. J. Dairy Sci. 104:4822–4836. https:// doi.org/10.3168/jds.2020-19084.

Bisutti, V., A. Vanzin, A. Toscano, S. Pegolo, D. Giannuzzi, F. Ta gliapietra, S. Schiavon, L. Gallo, E. Trevisi, R. Negrini, and A. Cecchinato. 2022. Impact of somatic cell count combined with dif ferential somatic cell count on milk protein fractions in Holstein cattle. J. Dairy Sci. 105:6447–6459. https://doi.org/10.3168/jds .2022-22071.

Comments 5: Lines 113-116: References. There are many publications that show much lower value of leukocytes account in milk without inflammation ~50% and ~50% epithelial cells

Response 5:

Thank you for your review comments. According to your guidance, we have revised the article. The revised parts are as follows and marked in red and yellow in the manuscript (line114~119):

“The somatic cells in milk are composed of 4 types of cells, namely polymorphonuclear leukocytes (PMN), lymphocytes (LYMs), and macrophages (MAC), and epithelial cells. In healthy mammary glands, SCC is generally low (<100,000 cells/mL) and macrophages represent the predominant cell type[20]. In infected udders, SCC increases and the proportion of different cells changes, with PMN reaching up to 95% of total SCC[21]. ”

Comments 6: Please remove all (SCC) apart from the first time change to SCC;

[116 somatic cell count (SCC); 130 SCC (somatic cell count) and 131 DSCC (differential somatic cell count)]

Response 6:

I am sorry that this was due to my negligence. The corresponding part has been revised in the manuscript and only the abbreviation has been retained (line 119,121, 133 and 153).

Comments 7: Lines 152 and 155. Please explain the numbers   23, 253 records of 6,841 lactating cows

Response 7:

During the data collection period, milk from healthy cows was collected and measured once a month to obtain indicators such as milk fat percentage and milk protein percentage and mid-infrared spectral data. A total of 6,481 lactating cows participated in the measurement. A cow would participate in the measurement for multiple months, so a cow would have multiple records. The total number of records was 23,253, which means that on average a cow participated in 3 to 4 measurements. Rather than measuring each cow once.

Comments 8: Lines 194.  Please explain “divided based on health diagnostic records of farms”

Health cow by the farmer with 350,000 SCC ????

Response 8:

We apologize for not describing this clearly in our previous manuscript.

As described in response 1, the health diagnosis of the farm means CMT screening and ELISA pathogen detection.

In addition, we did not mention 350,000 SCC as a healthy cow, so we guessed that this might be a misunderstanding.

Comments 9: Lines 324 and 332. When DSCC was > 50%, that is, PMN and LYM in somatic cells were dominant, milk yield significantly increased,

Check that it doesn't make sense.  That in an inflammatory state the cow will give more milk ????

Response 9:

Thank you for your valuable suggestions. We think there are several possible explanations, and they are written after the result analysis in the article (line 341~350, red font, yellow background):

“The possible explanation is that high-yielding dairy cows have larger milk production. When milking is incomplete (such as long milking intervals and loose nipple sphincters), milk is easy to accumulate and form a bacterial culture medium. Lactose and fat in milk provide nutrition for pathogens and promote infection. And the nipples of high-yielding dairy cows are in contact with milking equipment for a longer time, which makes them more susceptible to mechanical stimulation and damage, and the chance of bacterial invasion increases. In addition, high-yielding dairy cows have more serious negative energy balance, reduced immunity, and impaired mammary defense function. Then other components are in low concentration due to dilution effect.

Comments 10: Figure 1: please use Bar Graph when presenting data for different parity. This is average number not continuous.

Please provide SE during the text near each average number.

Response 10:

We agree with your comment and have presented the data related to parity in a bar chart in Figure 1(line 231).

And SE is added as mean ± SE (line 239~ 255, 263~269, 302, 317~319, 337~341, 356~358, 366~375, red font, yellow background).

Comments 11: Please submit separate graphs for all components for healthy and masitic cows

Response 11:

All components of healthy cows and cows with mastitis are described in the following table (LSMean±SE)

Traits

Healthy cow

Mastitis cow

P value

Milk yield, kg/d

31.53±0.46

29.18±0.45

0.039597

Fat, %

4.30±0.03

4.08±0.02

0.030529

Protein, %

3.39±0.01

3.35±0.01

0.046656

Total-casein, g/L

21.65±0.10

21.51±0.09

0.029586

αs1-CN, g/L

6.11±0.01

6.10±0.01

0.394033

β-CN, g/L

8.30±0.06

8.24±0.06

0.003424

κ-CN, g/L

3.80±0.03

3.76±0.03

0.003035

α-LA, g/L

0.68±0.02

0.65±0.02

5.22E-07

β-LG, g/L

7.45±0.07

7.55±0.08

0.019325

IgG, g/L

31.70±0.23

31.92±0.22

0.012286

LF, g/L

0.15±0.00

0.16±0.00

0.025399

Reviewer 2 Report

Comments and Suggestions for Authors

Dear authors,

The present document is about one of the world's healthy dairy problems. The objective was to show important data to increase knowledge about mastitis. However, some comments are needed to improve the paper.

Comments on the Quality of English Language

Sorry, I'm not authorized to assess English grammar and spelling. English is not my native language.

Author Response

Comments 1: What about of 305 -day production curve adjustment for all milk components????

Response 1:

Thank you for pointing this out. We have been mainly focusing on analysing protein fractions during our study and have been negligent of 305-day corrected milk yield, instead we have directly used the simple indicator of daily milk yield. In future studies, we will consider your suggestion to choose 305-day corrected milk yield or fat-protein-corrected milk yield as the research object to start the research on milk production traits of dairy cows.

Comments 2: Please, simplify the sentence.

The contents of α-LA, β-LG, and LF were all high at the early stage of lactation (0.64 g/L, 7.59 g/L, and 0.16 g/L, respectively), followed by a slight decrease at the peak lactation (0.59 g/L, 7.43 g/L, and 0.15 g/L, respectively), and then gradually increased and remained stable at the mid-to-late stage, reaching higher levels at the late stage (0.65 g/L, 8.62 g/L, and 0.20 g/L, respectively)

Response 2:

Agree. We simplify the sentence as follow (line248~251 in manuscript):

“The contents of α-LA, β-LG and LF were all higher in the early lactation, with the lowest value at the peak of lactation, and then increased with the increase of lactation period(from 0.64±0.02, 7.59±0.10, and 0.16±0.00g/L to 0.59±0.01, 7.43±0.10, and 0.15±0.00g/L and to 0.65±0.01, 8.62±0.10, and 0.20±0.00g/L, respectively)”.

And other similar sentences have also been adjusted.

Comments 2: Please, explain or describe the data of statistical analysis to get difference between healthy and mastitis group.

In the fig 2B, is imposible to see.

Response 2:

This is a good suggestion. According to the comments of reviewer 1, we have updated model 1 and the figure. The comparison in Figure 2B is now clearer than before. We have also added a description of the numerical values ​​in the manuscript just in case: “Milk yield, milk fat percentage, and milk protein percentage were significantly higher in the healthy group (31.53±0.46kg/d, 4.30±0.03%, 3.39±0.01%) than in the diseased group (29.18±0.45 kg/d, 4.08±0.02%, 3.35±0.01%) (P < 0.05, Figure 2A, B)”(line278~280)

Comments 3: Cavity?????? What do you refers???

Explain.

Response 3: We apologize for the ambiguity. We have re-investigated the cited reference and found that the original authors used the term alveoli. This may be due to negligence during the literature compilation, article writing and translation process. The original sentence in the literature is: During the milking process, the received milk fractions have a different composition, but it is not clear if this is due to a different passage of components through the BMB or different transport within the milk compartments. Milk somatic cells, mainly leukocytes that need to pass the BMB from blood to milk, are present in milk from healthy glands at different ratios in the different compartments. Polymorphonuclear granulocytes were shown to be the predominant cell type in the alveoli, whereas macrophages dominated the cells in the cistern of healthy quarters measured by fractionized milking. Most leukocytes traverse the epithelial barrier in the alveoli; however, they also pass the epithelial barrier of the teat cistern. The electrical conductivity is highest in the cisternal milk. Fat, lactose, and K concentrations are higher in the alveolar milk. Fat and lactose are released by the epithelial cells in the alveoli.

So we have revised the wording in the manuscript(line418). We apologize for the mistake again.

Comments 4: Please, this sentence is very important to considered, therefore, I suggest to put special attention about idea or sentece made in the page 8, lines 288-289 "certain change rule" I think that there is not enough balance data for thart.

Response 4: Agree. This is a very good suggestion. In the next step of this study, we will collect more detailed mastitis data to enrich the data set, study the changes in the ratio of PMN and LYM during mastitis, and further verify the results of this study.

Comments 5: Date of Ethical protocol is not in accordance with period of field sampling.

Response 5: Thanks for your reminder and sorry for misunderstanding again. The sample data measured by our team was not only used for the analysis of mastitis and milk protein components in this study, but earlier data was used to measure indicators such as fatty acids and minerals, and the indicators collected were continuously increased during the period. This study was carried out during the sampling project, not at the beginning.

Reviewer 3 Report

Comments and Suggestions for Authors

The mnauscript: 'The Relationship between Protein Fraction Contents and Immune Cells in Milk' is well prepared and focuses on important issue.

In my opinion, the M&M section could also include the information about herds size (and average).

Please, include better resolution figures - currently, they are not of the highest quality.

Minor comments:

  • lack of space before reference brackets - please, correct
  • 1-DSCC - please explain, above is only DSCC
  • lina 155 - lack of space - Dataset2(the second ....

Author Response

Comments 1: In my opinion, the M&M section could also include the information about herds size (and average).

Response 1:

Thank you for your good suggestion, which will make the Materials and Methods section more complete. We have added the relevant information to the manuscript: The average number of lactating cows on the farm was 650, ranging from 41 to 3527. (line150~151, yellow font and red background)

Comments 2:

Please, include better resolution figures - currently, they are not of the highest quality.

Response 2:

Thanks for your reminder, the figures had been reinserted.

Comments 3: Minor comments:

lack of space before reference brackets - please, correct

1-DSCC - please explain, above is only DSCC

lina 155 - lack of space - Dataset2(the second ....

Response 3:

Your carefulness and rigor are worthy of our learning. According to your suggestions, the corresponding positions in the manuscript have been modified. In the future research process, we will pay more attention to such issues.

Round 2

Reviewer 1 Report

Comments and Suggestions for Authors

Lines 53 and 334-335. “When DSCC increased to 50%, milk yield decreased”,

“When DSCC was > 50%, that is, PMN and LYM in somatic cells were dominant, milk  yield was higher, and milk fat percentage and milk protein percentage significantly decreased with increasing DSCC (P < 0.05)”. however in Fig. 4 A DSCC > 70% milk yield increased ????? Please explain

Lines 340-349. I suggest to delete your new adding  

Author Response

Comments 1: Lines 53 and 334-335. “When DSCC increased to 50%, milk yield decreased”,

“When DSCC was > 50%, that is, PMN and LYM in somatic cells were dominant, milk  yield was higher, and milk fat percentage and milk protein percentage significantly decreased with increasing DSCC (P < 0.05)”. however in Fig. 4 A DSCC > 70% milk yield increased ????? Please explain

Response 1: We sincerely appreciate the valuable comments. We have checked the literature carefully and expect to receive your approval. We believe that lines 425–428 of the manuscript may provide an explanation for this result, as outlined in Reference 30:

Given that the considered population was mostly clinically healthy, as also confirmed by SCS values, the LYM proportion in the DSCC is probably predominant with respect to PMN (Schwarz et al., 2011), so the high level of DSCC is not exactly mirroring an increased inflammation but rather a different cell type distribution. These findings support the importance of discriminating between leukocyte populations within DSCC as previously evidenced (Pegolo et al., 2021a, 2022a; Bisutti et al., 2022).

About 75% of the cows in dataset 2 had an SCC of less than 400*103 cells/mL, and the proportion of individuals with an SCC of less than 200*103 cells/mL was more than 50%. We therefore consider the description of Reference 30 to be applicable to the present results.

Reference 30:

ž   Giannuzzi, D., et al., Novel insights into the associations between immune cell population distribution in mammary glands and milk minerals in Holstein cows. Journal of Dairy Science, 2024. 107(1): p. 593-606. https://doi.org/10.3168/jds.2023-23729

Literatures mentioned in the citation:

ž   Schwarz, D., U. Diesterbeck, S. König, K. Brügemann, K. Schlez, M. Zschöck, W. Wolter, and C. Czerny. 2011. Flow cytometric dif ferential cell counts in milk for the evaluation of inflammatory reactions in clinically healthy and subclinically infected bovine mammary glands. J. Dairy Sci. 94:5033–5044. https://doi.org/10 .3168/jds.2011-4348.

ž   Pegolo, S., D. Giannuzzi, V. Bisutti, R. Tessari, M. Gelain, L. Gallo, S. Schiavon, F. Tagliapietra, E. Trevisi, P. Ajmone Marsan, G. Bit tante, and A. Cecchinato. 2021a. Associations between differential somatic cell count and milk yield, quality, and technological char acteristics in Holstein cows. J. Dairy Sci. 104:4822–4836. https:// doi.org/10.3168/jds.2020-19084.

ž   Pegolo, S., R. Tessari, V. Bisutti, A. Vanzin, D. Giannuzzi, M. Giane sella, A. Lisuzzo, E. Fiore, A. Barberio, E. Schiavon, E. Trevisi, F. Piccioli Cappelli, L. Gallo, P. Ruegg, R. Negrini, and A. Cecchi nato. 2022a. Quarter-level analyses of the associations among sub clinical intramammary infection and milk quality, udder health, and cheesemaking traits in Holstein cows. J. Dairy Sci. 105:3490 3507. https://doi.org/10.3168/jds.2021-21267.

ž   Bisutti, V., A. Vanzin, A. Toscano, S. Pegolo, D. Giannuzzi, F. Ta gliapietra, S. Schiavon, L. Gallo, E. Trevisi, R. Negrini, and A. Cecchinato. 2022. Impact of somatic cell count combined with dif ferential somatic cell count on milk protein fractions in Holstein cattle. J. Dairy Sci. 105:6447–6459. https://doi.org/10.3168/jds .2022-22071.

Comments 2: Lines 340-349. I suggest to delete your new adding  

Response 2: Thank you for the helpful comment. Upon reflection, we agree with you. Lines 340-349 have been deleted in the latest version.